# RobotKeyframing: Learning Locomotion with High-Level Objectives via Mixture of Dense and Sparse Rewards

**Fatemeh Zargarbashi**[1,2]    **Jin Cheng**[1]    **Dongho Kang**[1]    **Robert Sumner**[2]    **Stelian Coros**[1]

[1]ETH Zürich, Switzerland    [2]DisneyResearch Studios, Switzerland

fatemeh.zargarbashi@inf.ethz.ch

**Abstract:** This paper presents a novel learning-based control framework that uses keyframing to incorporate high-level objectives in natural locomotion for legged robots. These high-level objectives are specified as a variable number of partial or complete pose targets that are spaced arbitrarily in time. Our proposed framework utilizes a multi-critic reinforcement learning algorithm to effectively handle the mixture of dense and sparse rewards. Additionally, it employs a transformer-based encoder to accommodate a variable number of input targets, each associated with specific time-to-arrivals. Throughout simulation and hardware experiments, we demonstrate that our framework can effectively satisfy the target keyframe sequence at the required times. In the experiments, the multi-critic method significantly reduces the effort of hyperparameter tuning compared to the standard single-critic alternative. Moreover, the proposed transformer-based architecture enables robots to anticipate future goals, which results in quantitative improvements in their ability to reach their targets.

Project website: https://sites.google.com/view/robot-keyframing

**Keywords:** Legged Robots, Multi-Critic Reinforcement Learning, Transformer, Motion Imitation

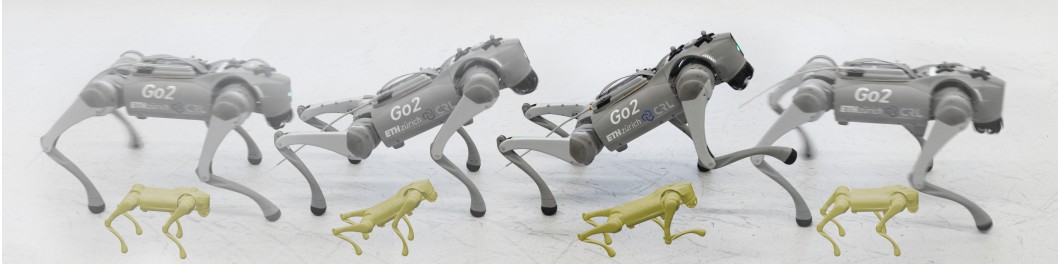

Figure 1: *RobotKeyframing*: Locomotion policy trained with our framework meets the keyframes with position and full posture targets (yellow) at specified times on hardware experiments.

## 1  Introduction

Legged robots hold a great promise for becoming household companions [1] or automated performers in the entertainment industry [2, 3]. In these applications, it is crucial for robot controllers to perform natural and directable behavior from simple high-level user command inputs beyond the typical commands used in the robotic domain such as joystick velocity commands [4, 5] or target base position [6, 7].

In the character animation domain, a widely used technique for specifying character behavior from simple and sparse inputs is *keyframing* [8, 9]. It involves defining the target position or kinematic

8th Conference on Robot Learning (CoRL 2024), Munich, Germany.

pose of the character at particular points in time, allowing animators to create smooth movements by interpolating between these keyframes. Despite its proven effectiveness within the kinematic animation pipeline, incorporating keyframing for achieving time-specific targets remains unexplored in the realm of physics-based robot control.

With inspiration from the character animation technique, we aim to equip legged robots with more refined control by incorporating sparse and temporal high-level objectives as keyframes. The primary goal of this work is to develop a locomotion controller that enables the robot to fulfill specified partial or full-pose targets while infilling natural behavior during the intermediate periods. This goal aligns with recent advancements in using reinforcement learning (RL) for legged robots due to their promising robustness and flexibility [10, 11]. However, learning a policy that accurately meets keyframes without imposing undesired constraints at intermediate periods presents challenges, particularly due to the need to handle sparsity in the keyframe objectives. Acquiring effective policies requires a meticulous reward design procedure that carefully balances these sparse rewards with other dense rewards which are crucial for regularizing and encouraging natural motion.

In this work, we present a novel framework that unifies timed high-level objectives with natural locomotion of legged robots through temporal keyframes. Along with the imitation objective similar to Peng et al. [12] for natural motion generation, our pipeline allows specifying full or partial high-level targets, including base position, orientation, and joint postures. We propose using a multi-critic RL framework to address the challenge of managing groups of sparse and dense rewards by learning distinct value functions. Our method also employs a novel transformer-based architecture to encode a variable number of goals with arbitrary time intervals. Unlike typical sequence-to-sequence transformers [13], we propose a lightweight sequence-to-token module that can be used autoregressively within a feedback control loop. We demonstrate the effectiveness of our framework through experiments both in simulation and on real-world hardware. Our policies successfully guide the robot to meet multiple keyframes at the required times, for both position and posture targets. Furthermore, the multi-critic approach showcases better convergence with less hyperparameter tuning compared to the conventional single-critic method. Our experiments also reveals that using a transformer-based encoder to anticipate future goals significantly enhances goal-reaching accuracy.

The contribution of this paper is threefold: (*i*) We introduce *RobotKeyframing*, a novel learning-based framework for integrating high-level objectives in natural locomotion of legged robots; (*ii*) We propose using multi-critic RL to effectively handle the mixture of dense and sparse rewards, along with a sequence-to-token encoder to accommodate a variable number of keyframes; (*iii*) We validate the effectiveness of our method through extensive experiments in simulation and on hardware.

## 2 Related Work

### 2.1 Reinforcement Learning for Legged Robots

Over the last decade, reinforcement learning has been increasingly applied to develop locomotion policies for legged robots [4, 14, 15]. The primary focus has been to achieve robust control policies that can accurately track velocity commands from joysticks [11, 16, 17]. More recently, researchers have attempted to enhance the versatility of legged robot controllers by incorporating high-level objectives, particularly through position- or orientation-based targets [18, 19, 20]. This high-level control is typically accomplished through hierarchical frameworks, where a high-level policy is learned to drive a low-level controller [7, 21, 22]. Conversely, end-to-end approaches aim to develop a unified policy for both high- and low-level control, allowing high-level objectives to directly influence low-level decisions [6, 18, 23]. However, the aforementioned methods typically urge the robot to reach a target as fast as possible, lacking refined control over the temporal profile of achieving the target. Inspired by keyframing in animation, this work aims to further expand control over robot motion by incorporating multiple keyframes as input to the control policy, thereby enabling robots to generate diverse behaviors in reaching targets. We further enhance this versatility by allowing partial or full targets, including base position, orientation, and joint postures.

## 2.2 Natural Motion for Characters and Robots

Synthesizing naturalistic behavior from existing motion datasets while fulfilling spatial or temporal conditions has been extensively studied in the character animation domain [24, 25, 26, 27]. Existing research for generating natural motions between keyframes [28, 29, 30] has mainly focused on the kinematic properties of characters and thus cannot be directly applied to physics-based characters or robots, whose dynamic interactions with the environment require consideration of both kinematics and dynamics. Various efforts have also been made to combine kinematic motion generation with physically controlled robots to achieve natural behavior on hardware [31, 32, 33, 34]. Another thread of research focuses on controlling characters in physically simulated environments, incorporating motion datasets as demonstrations [35, 36, 37, 38]. Some of these methods have also been successfully transferred to robot control for quadrupeds or humanoids [39, 40, 41, 42]. Among these works, Adversarial Motion Priors (AMP) [12] provides a flexible way to encourage the policy to have natural, expert-like behavior by connecting generative adversarial networks (GAN) [43] with RL given an offline motion dataset. We also incorporate an AMP-based imitation objective to encourage naturalistic motion for the policy and further extend it to infilling keyframes for robots.

# 3 Method

## 3.1 Problem Setup

To integrate high-level control objectives into the robotic control framework, we employ sparse keyframes that require a robot to achieve specific goals at predetermined times. Each keyframe contains a full or partial combination of a variety of targets such as global base position $\hat{\boldsymbol{p}} \in \mathbb{R}^3$, global base orientation $(\hat{\phi}, \hat{\zeta}, \hat{\psi}) \in \mathbb{R}^3$ where $\phi$, $\zeta$, $\psi$ denote roll, pitch, and yaw angles respectively, and full posture specified by joint angles $\hat{\boldsymbol{\theta}}_j \in \mathbb{R}^{N_j}$ where $N_j$ is the number of joints. Each keyframe is also assigned with a specific time $\hat{t} \in \mathbb{R}$ in the future at which the robot is expected to meet the goals. In summary, the high-level objectives are specified through these keyframes $\boldsymbol{K} = \left(\boldsymbol{k}^1, \boldsymbol{k}^2, ..., \boldsymbol{k}^{n_k}\right)$, where $\boldsymbol{k}^i = \left(\hat{\boldsymbol{g}}, \hat{t}\right)^i$ and $\hat{\boldsymbol{g}} \subset \left\{\hat{\boldsymbol{p}}, \hat{\phi}, \hat{\zeta}, \hat{\psi}, \hat{\boldsymbol{\theta}}_j\right\}$. Here, $n_k \leq N_k$ where $n_k$ and $N_k$ denote the actual and the maximal number of keyframes, respectively. We aim to support an arbitrary number of keyframes, allowing for the flexible specification of high-level objectives only as needed.

The main goal is to train a locomotion policy for legged robots that not only meets these keyframes but also maintains a natural style during the intervals between them. To avoid undesired restrictions on the intermediate periods, policy's task performance is evaluated exclusively at the designated times, making the keyframe objectives temporally sparse. However, relying solely on keyframes to train the control policy may result in undesirable motions. Thus, it is crucial to have additional rewards for regularizing and promoting a natural motion style. In this regard, we incorporate AMP [12] as a general style guide for the robot, encouraging the policy to behave naturally and similarly to an offline motion dataset from dogs [44]. The style and regularization rewards are evaluated at every step of the episode, making them temporally dense. The mixture of sparse and dense rewards presents a unique challenge that is difficult to manage effectively with standard RL frameworks. Details on the observation, action, reward definitions, and training setup can be found in Appendix A.

## 3.2 Multi-Critic RL for Dense-Sparse Reward Mixture

Modern RL algorithms [45, 46, 47] typically employ the actor-critic paradigm, where the actor decides the action to take, and the critic evaluates the action by estimating the value function. To effectively manage a complex mixture of temporally dense and sparse rewards, we employ a multi-critic (MuC) RL framework by Martinez-Piazuelo et al. [48] as shown in Fig. 2. It involves training a set of critic networks $\{V_{\phi_i}\}_{i=0}^n$ to learn distinct value functions associated with different reward groups $\{r_i\}_{i=0}^n$. Similar concepts have been used to balance a set of dense rewards [49, 50]; however, we extend the multi-critic method to the context of dense and sparse reward combination, where we demonstrate its distinct advantages. We design each reward group to contain either exclusively

dense or sparse rewards. This division is essential for effectively managing the distinct temporal characteristics of each reward type and facilitates value estimation.

We integrate the multi-critic concept to Proximal Policy Optimization (PPO) [46], as shown in Alg. 1. Particularly, each value network $V_{\phi_i}(\cdot)$ is trained independently for a specific reward group $r_i$ with temporal difference loss,

$$L(\phi_i) = \hat{\mathbb{E}}_t \left[ \| r_{i,t} + \gamma V_{\phi_i}(s_{t+1}) - V_{\phi_i}(s_t) \|^2 \right], \tag{1}$$

where $\hat{\mathbb{E}}_t$ is the empirical average and $\gamma$ is the discount factor. The value functions calculated by each critic are used to individually estimate the advantage $\{\hat{A}_i\}_{i=0}^n$ for each reward group. Subsequently, these advantages are synthesized into a policy improvement step by calculating the multi-critic advantage as a weighted sum of the normalized advantages from each reward group

$$\hat{A}_{MuC} = \sum_{i=0}^{n} w_i \cdot \frac{\hat{A}_i - \mu_{\hat{A}_i}}{\sigma_{\hat{A}_i}}, \tag{2}$$

where $\mu_{\hat{A}_i}$ and $\sigma_{\hat{A}_i}$ are the batch mean and standard deviation of the advantage from group $i$. Similar to PPO, the surrogate loss for policy gradient is clipped

$$L^{CLIP-MuC}(\theta) = \hat{\mathbb{E}}_t \left[ \min \left( \alpha_t(\theta) \hat{A}_{MuC,t}, \text{clip}(\alpha_t(\theta), 1 - \epsilon, 1 + \epsilon) \hat{A}_{MuC,t} \right) \right], \tag{3}$$

where $\alpha_t(\theta)$ and $\epsilon$ respectively denote the probability ratio and the clipping hyperparameter. This formulation integrates feedback from both dense and sparse rewards into the policy update, facilitating a balanced and effective learning process.

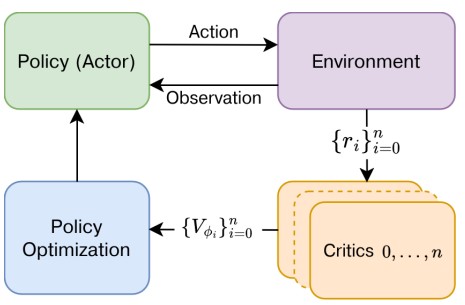

Figure 2: Multi-Critic RL.

**Algorithm 1** Multi-Critic PPO

1: Initialize policy parameters $\theta$ and parameters of each critic, $\phi_i$.
2: **for** $n = 1$ to $N$ **do**
3:     Rollout policy $\pi_\theta$ to fill the buffer.
4:     **for** each mini-batch **do**
5:         Estimate $\hat{A}_i$ for each $r_i$.
6:         Compute $\hat{A}_{MuC}$ with Eq. 2.
7:         Update policy with Eq. 3.
8:         Update each critic with Eq. 1.
9:     **end for**
10: **end for**

Assigning distinct critics for dense and sparse rewards helps achieve each set of objectives more effectively while reducing the reliance on extensive hyperparameter tuning. To illustrate this, consider a simple scenario with an episode length of $T$ involving two types of rewards: a temporally dense reward $r_d$ that is active at every step and a temporally sparse reward $r_s$ that is only active at the final step of an episode

$$r_{s,t} = \begin{cases} \hat{r}_s, & t = T \\ 0, & otherwise. \end{cases} \tag{4}$$

In the conventional single-critic RL, the total reward of each time step $t$ is typically computed as a linear combination of different reward terms $r_t = w_s r_{s,t} + w_d r_{d,t}$. The value in this scenario is

$$V(s_t) = \mathbb{E} \left[ w_s \gamma^{(T-t)} \hat{r}_s + w_d \sum_{k=t}^{T} \gamma^k r_{d,k} \right]. \tag{5}$$

We define the reward sparsity ratio as the number of dense reward steps per sparse reward horizon, which is here equal to $T$. The second term in Eq. 5 consists of a summation over $T - t$ individual reward terms, whereas the first term includes only a single component. This highlights the impact of different reward sparsities on the learning process, suggesting that the weight of reward groups must

be adjusted for different sparsity ratios to achieve a proper balance. This challenge is amplified when the sparsity ratio changes between episodes, for example, when keyframe timings are randomly sampled within a range. These variations can complicate the hyperparameter tuning process and hinder the efficacy of the learning algorithm.

In the multi-critic approach, on the other hand, the advantage for each reward group is normalized independently, ensuring that a fixed weight ratio for the advantages is adequate to maintain the desired balance, regardless of variations in the sparsity ratio. This method decouples reward frequency and magnitude from the learning process, enabling more effective policy optimization and reducing the effort for manual hyperparameter tuning.

### 3.3 Transformer-based Keyframe Encoding

The transformer framework [51] has achieved great success in modeling sequential data not only in the natural language processing [52, 53] but also in other areas including robotics [54]. The attention mechanism, serving as the core of transformer networks, models the correlation between each element of the input sequence and reweights them accordingly. To handle a variable number of keyframes in our problem, we utilize a transformer-based encoder to process the sequence of goals for both the policy and critics. However, unlike the typical application of transformers in sequence-to-sequence tasks, we adapt the architecture to function in a sequence-to-token manner, as shown in Fig. 3. This adaptation makes it suitable for autoregressive feedback control in robotic systems.

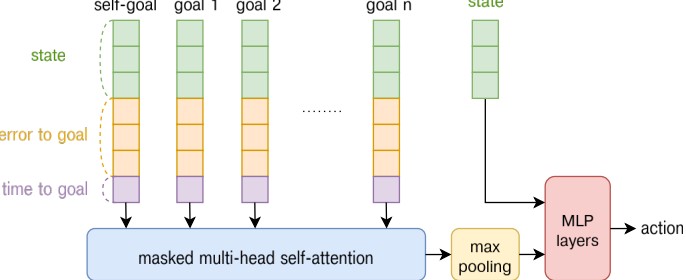

Figure 3: Policy with transformer-based keyframe encoder.

In our system, each input token $x_t^i$ is a vector corresponding to a particular keyframe. At every time step $t$, each keyframe $k^i$ is transformed spatially and temporally into a robot-centric view, resulting in a goal error $\Delta g_t^i$ and a calculated time to goal $\hat{t}^i - t$. These are then concatenated with the robot state $s_t$ to form a single token. Additionally, we incorporate a *self-goal* keyframe, $x_t^0$, as the first token in the sequence. This token represents a state with zero error and zero time to goal, which ensures that the control system remains operational despite the absence of active goals or after achieving all goals. The transformer encoder receives the sequence of tokens as a matrix $X_t = [x_t^0; \ldots; x_t^{n_k}]$, where $x_t^0 = [s_t, \mathbf{0}, 0]$, and $x_t^i = [s_t, \Delta g_t^i, \hat{t}^i - t]$ for $i = 1, \ldots, n_k$.

In scenarios where the number of active keyframes is less than the maximum capacity of the system, we apply masking to ignore the surplus tokens and focus only on the relevant keyframes. Furthermore, we also apply masking to keyframes once their designated time is reached and surpassed by a few steps. This practice prevents past goals from inappropriately influencing the long-term behavior of the policy. The output from the transformer encoder is then forwarded to a max-pooling layer, which condenses the encoded goal features for delivery to the subsequent multilayer perceptrons (MLP). By leveraging transformer's ability to handle sequences of varying lengths, our architecture can effectively integrate multiple and variable numbers of goals into the control process.

## 4 Results

The control policies are trained for quadruped robots with 12 degrees of freedom (DoF) using Isaac Gym [55]. At the start of each episode, the robot is either set to a default state or initialized according to a posture and height sampled from the dataset, a technique known as Reference State Initialization

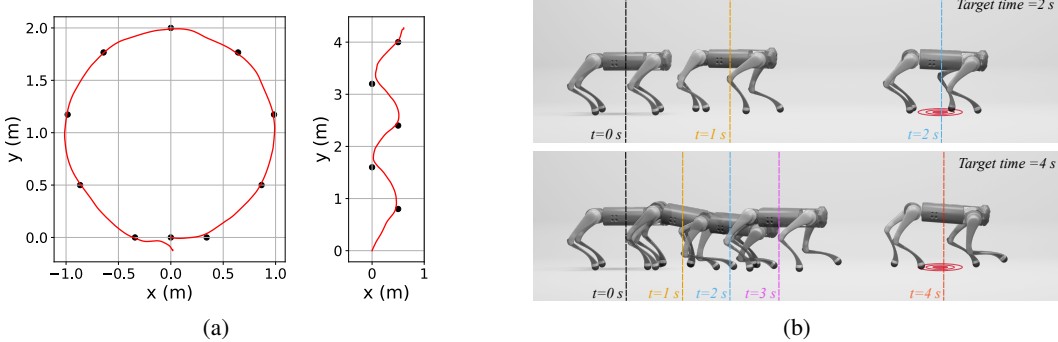

(a)

(b)

Figure 4: **a)** Horizontal trajectories of the robot base given two sets of position goals (dots). **b)** Specifying different temporal profiles generates diverse behaviors for the same position goal.

(RSI) [56]. We incorporate a learning curriculum, beginning with keyframes entirely sourced from reference data and progressively increasing the proportion of randomly generated keyframes, with time intervals, position targets, and yaw angles each sampled from a predetermined range. In this section, we present the qualitative and quantitative experiment results in simulation and on hardware.

## 4.1 Keyframe Matching

We demonstrate that our trained policy effectively reaches keyframes at the designated times through several simulation experiments. Given keyframes consisting of position goals, our policy reaches its targets with notable precision, as illustrated in Fig. 4a by the horizontal trajectories for two example scenarios with different number of keyframes. Furthermore, our framework offers control over target reaching time and can generate diverse behaviors for the same targets by specifying different time profiles. This is depicted in Fig. 4b through snapshots of robot motion when provided with keyframes consisting of the same position goal, but different target times. Full posture targets are also supported along with position and orientation goals. Fig. 5 shows snapshots of the robot motion given different keyframe scenarios, highlighting that our policy accurately meets its full posture targets while maintaining a natural style. Further quantitative evaluation of keyframe matching is provided in Appendix A.6.

## 4.2 Multi-Critic RL

In this section, we conduct a comparative analysis between multi-critic and single-critic approaches in the keyframing setup. Learning curves for both methods are presented in Fig. 6, with each method trained across three different ranges of sparsity ratios by sampling keyframes with varying time

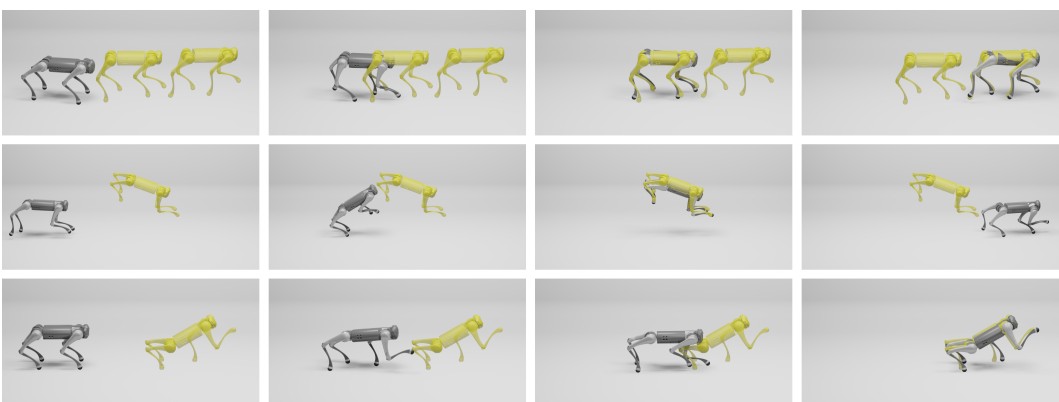

Figure 5: Snapshots of the robot motion given keyframes with full postures: moving forward (**top**), jumping (**middle**) and raising the paw up (**bottom**). Target keyframes are displayed in yellow.

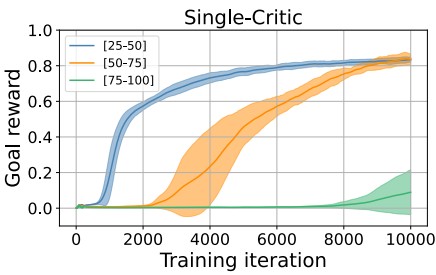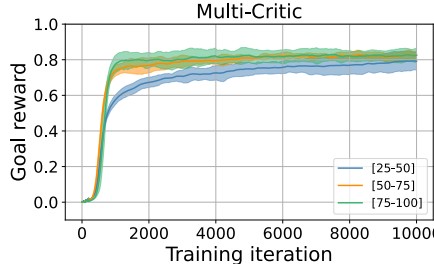

Figure 6: Convergence comparison of single-critic (**left**) and multi-critic (**right**) for different ranges of keyframe time horizons ($[25, 50]$, $[50, 75]$, $[75, 100]$) with fixed weights.

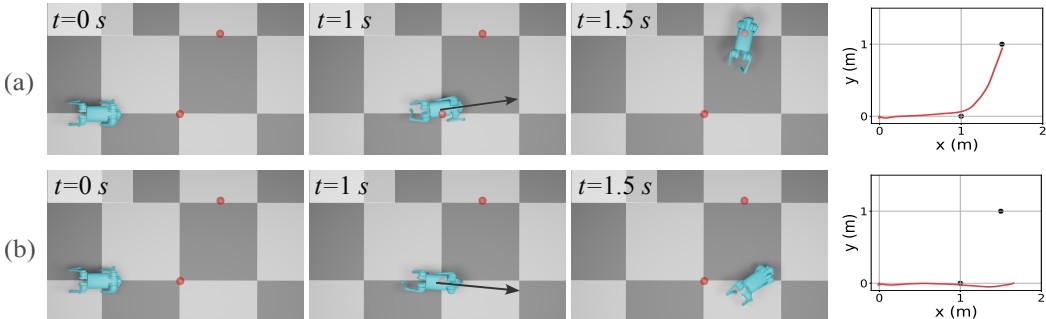

Figure 7: The policy aware of all goals (**a**) adjusts its yaw angle earlier to better reach the second goal compared to the policy only aware of the next goal (**b**). Keyframes are placed at 1 and 1.5 seconds in time. Left: snapshots, right: trajectories.

horizons. Initially, reward and advantage weights are tuned separately for single- and multi-critic according to the time horizon range $[25, 50]$. New policies are then trained using the same weights for another two scenarios of time horizons, $[50, 75]$ and $[75, 100]$. The learning curves reveal that the multi-critic algorithm achieves a similarly fast convergence without retuning the advantage weights for different scenarios. In contrast, the single-critic method displays significant delays in reward increase due to the sparser nature with longer keyframe horizons, underscoring the efficiency of the multi-critic in reducing the need for extensive manual hyperparameter tuning. This feature makes multi-critic particularly valuable in environments with varying reward sparsities.

### 4.3 Future Goal Anticipation

An advantage of using a transformer-based encoder is that it enables the policy to incorporate multiple and a varying number of goals as input. If the goals are temporally close to each other, awareness of future goals influences the robot's motion to achieve all of them more accurately. The phenomenon of future goal anticipation is demonstrated in Fig. 7 where we compare a policy aware of all goals and a policy only aware of the immediate next goal, both trained with only position goals in the keyframe. The policy trained with multiple keyframes adopts a larger yaw angle at the first goal, leaning more towards the second one to be able to reach it with higher accuracy. Table 1 provides a quantitative comparison of the two policies across three different scenarios: straight, turn and slow turn, the latter featuring a longer time horizon for the second goal. The results indicate that future goal anticipation helps the policy to adjust its motion while approaching earlier goals to gain better accuracy for the subsequent targets. This is particularly important when keyframes are temporally close, resulting in higher accuracy gains in fast and dynamic movements, compared to slower ones.

### 4.4 Hardware Deployment

We validate our method through extensive hardware experiments using the Unitree Go2 [57], a 12-DoF commercial quadruped robot. Fig. 8 illustrates the outcomes of a policy that manages up to 5 positional goals arranged in different courses, and a policy trained for full pose targets that

| First Goal | Straight | Turn | Turn (Slow) | |
|---|---|---|---|---|
| Aware of all goals | **0.0872 ± 0.0336** | **0.0781 ± 0.0236** | 0.0806 ± 0.0265 | 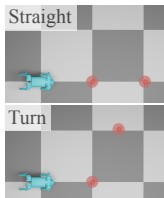 |
| Aware of next goal | 0.0898 ± 0.0317 | 0.0841 ± 0.0335 | **0.0787 ± 0.0208** | |
| **Second Goal** | Straight | Turn | Turn (Slow) | |
| Aware of all goals | **0.0472 ± 0.0187** | **0.3340 ± 0.1162** | **0.0566 ± 0.0804** | |
| Aware of next goal | 0.1332 ± 0.0605 | 0.7271 ± 0.1528 | 0.1711 ± 0.1071 | |

Table 1: Average position error (m) for three keyframe scenarios (depicted on right) across 20 experiments. The policy aware of all goals achieves better accuracy in reaching them.

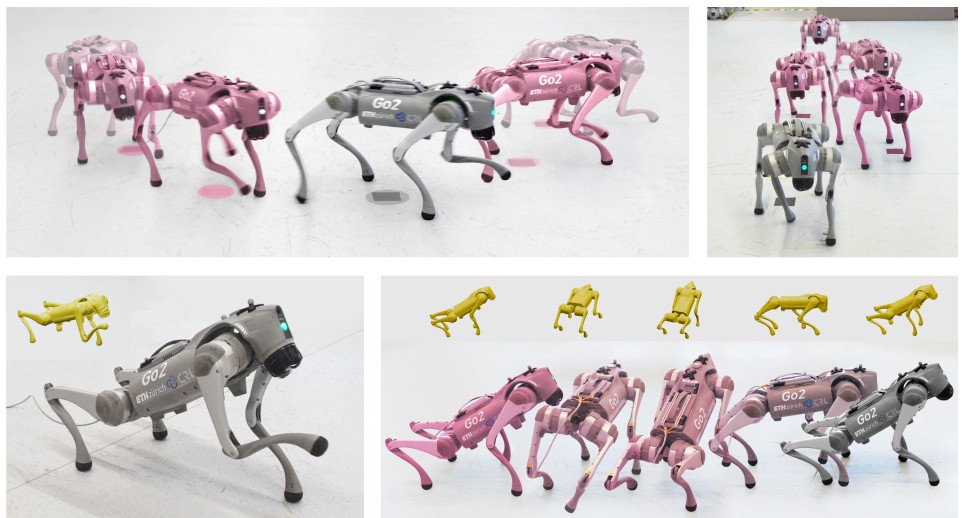

Figure 8: Hardware deployment of *RobotKeyframing* for position targets (**top**), and full-pose targets (**bottom**). Posture keyframes are displayed in yellow.

successfully drives the robot to achieve various posture keyframes. These experiments underscore the adaptability and effectiveness of our keyframing approach in enhancing high-level control in robotic systems. Readers are encouraged to watch the videos provided in the supplementary material for a more comprehensive presentation of these results.

## 5 Discussion

**Conclusion**: This paper presents *RobotKeyframing*, a learning-based control framework designed to incorporate high-level objectives into the natural locomotion of legged robots through a sequence of keyframes, leveraging transformer models and multi-critic reinforcement learning. Through simulation and hardware experiments, we demonstrated the effectiveness of our approach: The sparse reward imposed by keyframe objectives is effectively handled by a multi-critic PPO algorithm. In addition, the transformer-based architecture is adaptive to various number of target keyframes and improves accuracy in reaching targets through future goal anticipation.

**Limitations and future work**: First, if the timing values are infeasible for the specified goals, the robot may fail to meet the targets. However, it is worth noting that such cases do not result in uncontrolled behaviors, such as falling down. Second, our approach inherits the mode collapse issue from the AMP framework [12], which can be mitigated in future research through the integration of style embeddings. Third, the performance of our policy is currently limited by the motions in the dataset, restricting its ability to generalize to out of distribution motions or targets. Looking ahead, our method can be expanded to incorporate diverse types of goals in the keyframes, such as end-effector targets or more intuitive high-level inputs such as skill or text. Additionally, *RobotKeyframing* can be extended to more complex characters and potentially used for physics-based motion in-betweening in the character animation domain.

**Acknowledgments**

The authors would like to thank Miguel Zamora for valuable discussions. This work was supported by the European Research Council (ERC) under the European Union's Horizon 2020 research and innovation program (grant agreement No. 866480).

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

# A Appendix

## A.1 Observation and Action Space

The observation of the policy is composed of two main components: state observation and goal observation. State observation at time $t$ include the linear velocity ($\boldsymbol{v}$) and angular velocity ($\boldsymbol{\omega}$) of the base in local coordinates, current joint angles ($\boldsymbol{\theta}_j$), current joint velocities ($\dot{\boldsymbol{\theta}}_j$), projected gravity in the base frame ($\boldsymbol{g}_{proj}$), base height ($h$) and previous actions from the last timestep ($\boldsymbol{a}_{prev}$),

$$\boldsymbol{s}_t = \{\boldsymbol{v}, \boldsymbol{\omega}, \boldsymbol{\theta}_j, \dot{\boldsymbol{\theta}}_j, \boldsymbol{g}_{proj}, h, \boldsymbol{a}_{prev}\}_t. \tag{6}$$

A variable number of keyframes $\boldsymbol{K} = \left(\boldsymbol{k}^1, \boldsymbol{k}^2, ..., \boldsymbol{k}^{n_k}\right)$ are specified as targets for the robot. At each time step $t$, each keyframe $\boldsymbol{k}^i$ is transformed spatially and temporally into a robot-centric view. Then, the goal observation is prepared by calculating the remaining time to goal $\hat{t}^i - t$ and the error to target goals ($\Delta \boldsymbol{g}_t^i$),

$$\Delta \boldsymbol{g}_t^i \subset \{\Delta \boldsymbol{p}_b^i, \Delta \phi^i, \Delta \zeta^i, \Delta \psi^i, \Delta \boldsymbol{\theta}_j^i\}. \tag{7}$$

Here, $\Delta \boldsymbol{p}_b^i$ denotes the error between robot base position and keyframe position in the base coordinate frame, $\Delta \boldsymbol{\theta}_j^i$ is the error in joint angles, and $\Delta \phi^i$, $\Delta \zeta^i$ and $\Delta \psi^i$ denote the errors in roll, pitch and yaw angles, respectively, which are wrapped to $(-\pi, \pi]$.

The policy receives the sequence of tokens $\boldsymbol{X}_t = (\boldsymbol{x}_t^0, ..., \boldsymbol{x}_t^{n_k})$ as input to the encoder, where $\boldsymbol{x}_t^0 = (\boldsymbol{s}_t, \boldsymbol{0}, 0)$, and $\boldsymbol{x}_t^i = (\boldsymbol{s}_t, \Delta \boldsymbol{g}_t^i, \hat{t}^i - t)$ for $i = 1, ..., n_k$. Thanks to the transformer-based keyframe encoding, the extra tokens can be masked to enable arbitrary number of goals. In addition, keyframes with a time over one second past the current time are also masked to avoid any long-term influence on reaching the future goals.

The action ($\boldsymbol{a}_t$) space of the policy is set to target joint angles, which are tracked using a PD controller to compute the motor torques.

## A.2 Reward Terms

We include three groups of rewards in this framework: regularization, style, and goal. For each reward group, the final reward is computed as a multiplication of individual reward terms,

$$r_{\text{group}} = \prod_{i \in \text{group}} r_i. \tag{8}$$

Regularization rewards are designed to provide a smooth output of the policy and consist of several terms defined in Table A1. Here, $\mathcal{K}$ is an exponential kernel function defined in Eq. 9 where $\sigma$ and $\delta$ are the sensitivity and tolerance of the kernel function, respectively.

$$\mathcal{K}(\mathbf{x}, \sigma, \delta) = \exp\left(-\left(\frac{\max(0, \|\mathbf{x}\| - \delta)}{\sigma}\right)^2\right) \tag{9}$$

To generate natural motion between the keyframes, we use AMP proposed by Peng et al. [12], which involves training a discriminator $\mathcal{D}$ to identify motions that are similar to those of the offline expert dataset. The style reward is defined based on the discriminator output of the latest state transition of the robot ($\mathbf{s}_{t-1}, \mathbf{s}_t$),

$$r_{\text{style}} = \max\left(1 - 0.25(\mathcal{D}(\mathbf{s}_{t-1}, \mathbf{s}_t) - 1)^2, 0\right). \tag{10}$$

Table A1: Regularization reward terms

| | |
|---|---|
| Action rate | $\mathcal{K}(\dot{\mathbf{a}}, 8.0, 0)$ |
| Base horizontal acceleration | $\mathcal{K}(\ddot{\mathbf{p}}_{xy}, 8.0, 0)$ |
| Joint acceleration | $\mathcal{K}(\ddot{\boldsymbol{\theta}}_j, 150.0, 10.0)$ |
| Joint soft limits | $\mathcal{K}\left(\max\left(\boldsymbol{\theta}_j - \boldsymbol{\theta}_{j,max}, \boldsymbol{\theta}_{j,min} - \boldsymbol{\theta}_j, \boldsymbol{0}\right), 0.1, 0\right)$ |

Table A2: Goal reward terms

| Goal position | $\Phi^i(\mathcal{K}(\boldsymbol{p} - \hat{\boldsymbol{p}}^i, 0.2, 0))$ |
|---|---|
| Goal roll | $\Phi^i(\mathcal{K}(\phi - \hat{\phi}^i, 0.1, 0))$ |
| Goal pitch | $\Phi^i(\mathcal{K}(\zeta - \hat{\zeta}^i, 0.1, 0))$ |
| Goal yaw | $\Phi^i(\mathcal{K}(\psi - \hat{\psi}^i, 0.3, 0))$ |
| Goal posture | $\Phi^i(\mathcal{K}(\boldsymbol{\theta}_j - \hat{\boldsymbol{\theta}}_j^i, 0.2\sqrt{12}, 0))$ |

Goal rewards are defined with a temporally sparse kernel $\Phi^i(x)$

$$\Phi^i(x) = \begin{cases} x, & t = \hat{t}^i \\ 0, & \text{otherwise} \end{cases}, \tag{11}$$

and only activated when the corresponding timestep for that goal $\hat{t}^i$ is reached in the episode. The detailed reward terms are defined in table A2.

For advantage weights (Eq. 2), we recommend starting with a 1:1 ratio and making minor adjustments based on specific preferences. In our experiments, we used $\omega_{style} = 0.5, \omega_{goal} = 0.5, \omega_{reg} = 0.1$, maintaining equal weights for goal and style, with a lower weight for the regularization critic.

### A.3 Dataset Preparation

We use a database of motion capture from dogs introduced by Zhang et al. [44]. The motions are retargeted to the robot skeleton using inverse kinematics for the end-effectors' positions with some local offsets to compensate for the different proportions of the robot and dog. A subset of around 20 minutes of data was used, removing the undesired motions such as smelling the ground, walking on slopes, etc. We augment this dataset with other motion clips animated by artists to include more diversity in the dataset. The frame rate is adjusted to that of the simulation, i.e. 50 frames per second. In addition, the linear and angular velocity of the base are precomputed for each frame using a finite difference method and stored in the dataset.

### A.4 Training Procedure

We utilize Isaac Gym [55] for simulating the physical environment. At the start of each episode, the robot is initialized either to a default state (20% probability) or using Reference State Initialization (RSI) (80% probability). For RSI, a random frame from the reference motion trajectory provides the height, orientation, joint angles, base linear velocity, base angular velocity, and joint velocities for initializing the robot. RSI plays a crucial role in capturing and learning the specific style of motion, as highlighted in previous studies such as Peng et al. [56].

We employ two keyframe sampling strategies: dataset-based and random sampling. In the dataset-based approach, a motion trajectory and starting frame index are initially sampled. Then the time interval between keyframes are uniformly drawn from a specified range. The corresponding frames from the dataset are used to obtain the global position, orientation, and posture of the robot at the keyframe times. The global horizontal positions are subsequently converted to relative local positions with respect to the starting frame and then transformed into positions in the simulator's world coordinates by accounting for the robot's initial state. The yaw angle is similarly adjusted, while height, roll, pitch, and joint postures are directly used as target values.

In the random sampling strategy, time intervals are sampled from a predefined range, similar to the first strategy. The radius and direction of each keyframe, relative to the previous one are uniformly sampled to compute the keyframe's global horizontal position. Additionally, the change in yaw angle is sampled and added to the previous keyframe's yaw. The roll and pitch angles, joint postures and height are taken directly from a reference frame randomly selected from the dataset. This ensures that the posture is feasible and compatible with the height. Details of the sampling ranges are provided in Table A3.

Table A3: Keyframe sampling ranges

| Quantity | Time interval (steps) | Waypoint radius (m) | Waypoint direction (rad) | Delta yaw angle (rad) |
|---|---|---|---|---|
| Range | $[25, 50]$ | $[0.5, 1.0]$ | $[-\pi/3, \pi/3]$ | $[-\pi/3, \pi/3]$ |

Our methodology incorporates a learning curriculum, beginning with keyframes entirely sourced from reference data and progressively increasing the proportion of randomly generated keyframes. The meticulous sampling of target keyframes is critical for ensuring their feasibility and preventing them from impeding effective policy learning.

We train the policy to handle a maximum number of keyframes, randomly selecting the actual number of keyframes for each episode. To avoid negative impacts on training, unused goals are masked when input into the transformer encoder. For stability, the episode does not terminate immediately after the last goal is reached; instead, it terminates approximately one second later. The training setup for a full keyframe comprising time, position, roll, pitch, yaw, and posture targets with up to 5 maximum keyframes requires approximately 17 hours on a system equipped with Nvidia GeForce RTX 4090.

## A.5 Hardware Implementation Details

Domain randomization is added during training to achieve a robust policy that can be executed on hardware. Similar to Kang et al. [58], we randomize friction coefficients, motor stiffness and damping gains and actuator latency. Furthermore, we add external pushes during training. Details of domain randomization ranges are provided in Table A4. Although joint limits are softly taken into account in the simulation, we found it crucial to terminate episodes when reaching joint limits to ensure a stable deployment on hardware. We use a motion capture system to receive the global position and orientation of the robot. These are used to compute the relative errors to the target goals and are then passed to the policy. Other observations are computed based on the outputs from the state estimator. Our locomotion control policy runs at 50 Hz and updates the joint angle targets. At a lower-level, the robot's built-in motor controller operates at 1000 Hz, updating the torque targets for each joint. To enhance computational efficiency during RL training, we simulate the robot and the PD controller at 200 Hz, which has proven effective in practice.

Table A4: Domain randomization ranges

| Quantity | Range |
|---|---|
| Friction Coefficient | $[0.25, 1.75]$ |
| Push velocity ($m/s$) | $[0.0, 1.0]$ |
| Push angular velocity ($m/s$) | $[0.0, 1.0]$ |
| Stiffness multiplier | $[0.9, 1.1]$ |
| Damping multiplier | $[0.9, 1.1]$ |
| Actuator lag steps | $[0, 6]$ |

## A.6 Keyframe Matching

We provide quantitative analysis of keyframe matching performance for the three scenarios from Fig. 5 with full-pose keyframes in Table A5. The mean and standard deviation of each entry is from 20 independent evaluations. The posture error reported is the root mean squared error (RMSE) of all joint angles. The results indicate that our trained policy tracks the keyframes with good precision and consistency.

To assess the generalization capability of our method, we evaluated the trained policy using three different keyframe sampling strategies:

Table A5: Keyframe matching error for three different scenarios

| Scenario | Walk (goal 1) | Walk (goal 2) | Jump | Paw up |
|---|---|---|---|---|
| Distance error (m) | $0.048 \pm 0.011$ | $0.020 \pm 0.003$ | $0.118 \pm 0.012$ | $0.044 \pm 0.015$ |
| Roll error (deg) | $3.094 \pm 0.327$ | $1.621 \pm 0.074$ | $1.473 \pm 0.579$ | $1.209 \pm 0.733$ |
| Pitch error (deg) | $0.332 \pm 0.178$ | $0.934 \pm 0.235$ | $1.558 \pm 0.940$ | $0.516 \pm 0.246$ |
| Yaw error (deg) | $1.209 \pm 0.384$ | $2.120 \pm 0.367$ | $7.110 \pm 1.060$ | $2.773 \pm 1.301$ |
| Posture error (deg) | $2.469 \pm 0.309$ | $2.595 \pm 0.126$ | $3.226 \pm 0.103$ | $4.698 \pm 0.808$ |

- **Dataset**: Sampling the full pose (position, orientation, and posture) only from trajectories in the reference dataset.
- **Random**: Sampling the posture from the reference dataset while randomly sampling the planar position and yaw angle targets.
- **Evaluation dataset**: Sampling the full pose from the evaluation dataset, which was not seen by the policy during training. For the evaluation dataset, we use a few locomotion clips from the open-sourced dataset from Han et al. [59].

The mean and standard deviation for each strategy are derived from 20 independent evaluations and shown in Table A6. The results indicate that our policy generalizes outside the trained dataset in terms of position and yaw angle when exposed to the randomly sampled target during training. Surprisingly, the policy achieves decent keyframe tracking in position and orientation as well as postures with the evaluation dataset which is never seen during training. Although Table A6 shows promising generalization results, full generalization to postures that differ significantly from those in the training dataset is yet to be achieved.

Table A6: Keyframe matching performance across different keyframe sampling strategies

| Keyframes source | Dataset | Random | Evaluation dataset |
|---|---|---|---|
| Distance error (m) | $0.060 \pm 0.050$ | $0.142 \pm 0.087$ | $0.064 \pm 0.060$ |
| Roll error (deg) | $1.873 \pm 1.888$ | $3.181 \pm 3.466$ | $3.231 \pm 9.053$ |
| Pitch error (deg) | $3.084 \pm 6.747$ | $4.260 \pm 9.202$ | $1.438 \pm 2.842$ |
| Yaw error (deg) | $2.394 \pm 1.775$ | $8.754 \pm 6.220$ | $2.200 \pm 2.235$ |
| Posture error (deg) | $5.119 \pm 3.143$ | $4.640 \pm 1.606$ | $9.207 \pm 7.122$ |

## A.7 Future Goal Anticipation

Details of target keyframes used for Table 1 are given in Table A7.

Table A7: Details of keyframe scenarios

| Scenario | First Goal | | Second Gaol | |
|---|---|---|---|---|
| | Time (steps) | Position (m) | Time (steps) | Position (m) |
| Straight | 50 | $(0, 0.32, 1.0)$ | 75 | $(0, 0.32, 2.0)$ |
| Turn | 50 | $(0, 0.32, 1.0)$ | 75 | $(1.0, 0.32, 1.5)$ |
| Turn (Slow) | 50 | $(0, 0.32, 1.0)$ | 100 | $(1.0, 0.32, 1.5)$ |

## A.8 Training Hyperparameters

Table A8 provides details of hyperparameters used for training.

Table A8: Summery of training hyperparameters

| | |
|---|---|
| Number of environments | 4096 |
| Number of mini-batches | 4 |
| Number of learning epochs | 5 |
| Learning rate | 0.0001 |
| Entropy coefficient | 0.02 |
| Target KL divergence | 0.02 |
| Gamma | 0.99 |
| Lambda | 0.95 |
| Discriminator learning rate | 0.0003 |
| Discriminator update rate per epoch | 20 |
| Discriminator batch size | 768 |
| Transformer encoder layers | 2 |
| Transformer heads | 1 |
| Transformer feed-forward dimensions | 512 |
| MLP dimensions | $[512, 256]$ |
| Initial standard deviation | 1.0 |
| Activation function | ELU |

