# OpenReview forum: "RobotKeyframing: Learning Locomotion with High-Level Objectives via Mixture of Dense and Sparse Rewards"
_robot-learning.org/CoRL/2024/Conference — CoRL 2024_

### Official Review · Reviewer_fZ8e · 2024-07-19

**Originality:** 3
**Technical Quality:** 3
**Clarity Of Presentation:** 3
**Potential Impact:** 3
**Recommendation:** 2
**Confidence:** 5

**Review:**

While the technical contribution is quite limited, the proposed multi-critic approach could potentially have impact by allowing to learn policies that follow more expressive commands, which is important for hierarchical control.

The paper is well-written and easy to follow. However, the pipeline is not described in sufficient detail. In its current form, the paper is not reproducible. Providing source code would be very helpful for filling the blanks. Please refer to my questions below for examples. In particular, I am wondering how the keyframes are tokenized and sampled and how exactly the initial states for the RSI are sampled.

The main strength of the paper are the good empirical results. However, the paper does not provide quantitative results on the hardware, but only shows some videos of successful attempts few (I assume to some extend tuned) keyframe-scenarios. The paper would be much stronger, when using held-out demonstrations (ideally from a different dataset) to sample keyframes and quantitatively evaluating their reward in simulation and on the robot.

**Quality Of The Limitations Section:**

3

**Questions For Rebuttal:**

How exactly are the keyframe tokens passed to the transformer? A token typically refers to a single integer, however, I assume that the state, relative goal, and time-delta are not discretized into a single integer.

The appendix provides some information on sampling the keyframes. However, I think it would be useful to describe the process in more detail.

It would also be useful to describe the sampling of the initial state (with RSI) in more detail, e.g. by providing code or pseudo-code.

Appendix states that the "previous actions" are part of the state. Does it refer to the actions of the last time step, or a longer history of actions?

I assume the loss in Eq. 1 uses a target network. It would be could to indicate this in the equation.

What are the "time steps" in the x-axis of Fig. 6? Are these RL iterations or is there missing some factor to bring them to the right order of magnitude?

According to the appendix, the RL policy runs at 50Hz. What are the control frequencies of the PD controller?

The joint soft limits (Table A1) seem to bias the joint position towards the middle position, $1/2 (\boldsymbol{\theta}\_{j,min} - \boldsymbol{\theta}\_{j,max})$. Does this joint configuration correspond to a reasonable nominal posture for the robot?

How often is the discriminator updated during RL?

**Robotics Focus:**

4

**Summary Of Paper:**

The paper presents a reinforcement learning pipeline for learning for learning quadruped locomotion policies in simulation and demonstrates transfer to a Unitree Go2. Whereas prior methods are often restricted to velocity commands, the proposed method can set a variable number of keyframes consisting of timestamps, target poses and joint configurations. The keyframe reward, which is sparse as it is only given at the respective timestamps, is complemented by an AMP style reward obtained from the popular dog motion capture dataset by Zhang et al. 2018. The main contribution is the use of a multi-critic approach, that rather than trading-off the different reward terms (style reward and keyframe reward) independently estimates the advantage functions for the these rewards, normalizes them, and weights them for trading-off. While this multi-critic approach is not novel, the idea is relatively novel, hasn't been applied to locomotion and seems interesting, because in contrast to reward-weights, the weights on the advantage functions work well for different degrees of sparsity (timegaps between keyframes). The paper presents nice empirical results, as the quadruped performs natural locomotion while reaching the desired keyframes at the request timestamps.

**Summary Of Recommendation:**

Good work with sufficient relevance to the field but insufficient description of the methodology.

---

### Official Review · Reviewer_i8Sh · 2024-07-20
**Good execution for a new task/framework**

**Originality:** 3
**Technical Quality:** 4
**Clarity Of Presentation:** 4
**Potential Impact:** 3
**Recommendation:** 3
**Confidence:** 5

**Review:**

Overall I think the work is fairly strong. The novelty is relatively limited and is more of a system putting together different previous researched techniques (with some modifications), but I think is moving towards a good direction for pose matching. I usually see works more along the lines of either matching whole trajectories or “style” based methods, but a keyframe based approach as shown here so that behaviors can be specified in an animation like style is more convenient than whole trajectories but more precise than style methods. This work is a good first step towards this, and if more improvements can be made in the future I think this can be a useful framework for control.

## Strengths:

The proposed method can achieve (at least visually) very good matching of keyframes, even for those more difficult ones when the robot is not in contact with the ground like jumping. This can allow for very easy specification of detailed behaviors. Being able to command behaviors through an animation like keyframing can be very useful. The system is a very sound combination of techniques and has some good ideas.

I particularly like the separation of sparse vs. dense rewards, which is a good application of multi objective/critic techniques and I think is very helpful. The simple example given at the end of section 3.2 is a good illustration of the need for something like this when there are rewards that can only be achieved a few times per episode.

The text and method are for the most part clear and understandable. Explanation details are mostly thorough and go into the sufficient greater detail of the training in the appendix.


The technical details of the methods and analysis of the algorithm seem to be all correct to my understanding.


## Limitations:

The overall novelty of the proposed method is a little limited. While I don’t think they have been put together for this specific keyframe matching task, and have slight variations, the basic idea of each proposed technique has been done before.
The multi-ciritic paradigm is just multi-objective RL and the method used here is pretty much the same as those used in the works that you reference (48, 49, 50), just that you have a “special” choice of what your reward groups are (dense vs. sparse).
For example, Radosavovic et. al [1] also use a sequence-to-token transformer architecture for their policy (current and previous observations are the tokens in their case rather than goals) in an autoregressive manner.
I’m not really sure if both of these points are really novel and thus if point ii (line 55 in the text) is really a contribution. I think the contribution here is really the system as a whole used for this keyframe matching task, i.e. contribution i.

I would like to see some evaluation of the actual keyframe matching. The only data I see for the goal error is Table 1, and this seems to be for just a body position keyframe. I would like to see how accurately the learned policy can match given full pose keyframes like those shown in Figure 5. Another table similar to Table 1, but instead reports average pose error (maybe need to do some scaling so joint angles and body pose/orientation are on the same scale?) for the keyframe scenarios in Figure 5 would be good to add. I think this is pretty important since keyframe matching is the main purpose and goal of your system.

I would maybe consider changing the title (if possible) to not include “locomotion”, since I think that locomotion is not really a part of your goal and the purpose of the proposed method is for matching keyframes instead. Of course you can keyframe a locomotion gait, but I think this is probably not a very convenient way to specify locomotion behaviors and the results are less performant, robust, and natural looking than all of the previous works (which you reference too, [5, 7, 10-12, 14, 15, etc.]). The main purpose and novel part that you are proposing is more general in that you can match arbitrary poses.

The limitation that is mentioned in Section 5 that the keyframe poses is limited to motions in the dataset is a pretty large limitation. As it is right now, I don’t think this system has an actual use case over just AMP. However I do understand that this is a first work for this method, and that future works can have improvements and perhaps have more generalization that would put this above trajectory imitation works like AMP.

Video comments:
Can you use some sort of texture for the floor in the simulation results? The pure white floor makes it hard to judge where the floor actually is and when the feet are actually in contact with it or not.


[1] Ilija Radosavovic et al. ,Real-world humanoid locomotion with reinforcement learning. Sci. Robot. 9,eadi9579(2024). DOI:10.1126/scirobotics.adi9579

**Quality Of The Limitations Section:**

3

**Questions For Rebuttal:**

I don’t fully get the use of a transformer-based architecture since it can’t really handle arbitrary number of keyframes as used here. There is still a fixed maximum number of keyframe goals it can handle and masking is still required. How important is the self attention for this system? If you need to have masking anyway, why can you not just individually encode these keyframes and pass them directly into the MLP? Why not have some positional encoding of each keyframe so that the policy can handle any number of keyframe goal inputs?

Why does the state need to be part of goal tokens X_t? It seems like the state is the same across each goal token, and the state is already inputted to the MLP layers. What is the intuition for why the state should influence which goal to pay attention to? How does the policy perform if the state is not part of the goal tokens?

Since there is the self-goal keyframe which has 0 error and 0 time to goal, how does the policy differentiate between this self-goal keyframe and a masked keyframe (which I assume is also just all zeros)? I guess in the self-goal case the state will not masked, is this the purpose of including the state in the goal keyframes?

You mention that past goal keyframes are masked as well, are they just masked in their current place or are the next goal keyframes “moved up”? For example, if at the start the goal tokens are inputted like $(x^0_t, x^1_t, …, x^{n_k}_t)$ then some time later (let’s say at $t = w$) after 2 goals have been reached are goal tokens like $(0, 0, x^2_w, …, x^{n_k}_w)$ or would it be like $(x^2_w, …, x^{n_k}_w, 0, 0)$? The text sounds like it does the first option, but I think the second option makes more sense? Along the same line, is the self-goal keyframe ever masked? Or will it always be a part of the input?

How important is being aware of all next goals rather than just the next two? The evaluation provided (Table 1) only tests when there are 2 keyframes, not when there are more than 2. I can understand that being aware of more than just the next goal is useful so that the policy can prepare (as Table 1 shows), but how far in the future is it necessary to look ahead? If for example there are 3 keyframe goals, is it necessary to know all 3 or can they just be given two at a time (1st and the 2nd, and then the 2nd and 3rd after the 1st goal is past). I think this is an important evaluation to do, since it answers if handling a variable number of keyframes (a large part of your method) is even necessary or just a small fixed lookahead works for all cases.

What are the weightings ($w_i$) to used to weight the sparse vs. dense advantages? How is this hyperparameter tuned?

**Robotics Focus:**

4

**Summary Of Paper:**

This paper presents a method for learning a “keyframe” policy that can satisfy a sequence of keyframe poses. This is done through a transformer based architecture to handle a variable number of goal keyframes as input, along with a multi-critic RL process to more evenly weight dense vs. the sparse keyframe matching rewards. There is also flexibility in what the keyframes specify, allowing for any full or partial combination of base pose and joint angles. Simulation results are shown tracking both position waypoint goals along with full posture key frames for walking, jumping, and paw raising motions. Ablation studies vs. a single critic method are presented, showing the robustness of using multiple critics with varying reward sparsities. Finally hardware results are shown of the system running on a Unitree Go2 quadruped doing locomotion and posture matching tasks.

**Summary Of Recommendation:**

The novelty is a bit limited but it is moving towards a good direction. I think that this keyframe approach can have some advantages over the style based methods that are more commonly used today. Full generalization has yet to be shown in this work but it is a first step towards some thing that could do general pose matching, which would be pretty powerful.

---

### Official Review · Reviewer_aL5k · 2024-07-20
**Clear paper, strong contribution**

**Originality:** 4
**Technical Quality:** 4
**Clarity Of Presentation:** 5
**Potential Impact:** 4
**Recommendation:** 4
**Confidence:** 4

**Review:**

Quality

High-quality, clean, well-written paper, with clear and well-presented idea. The authors have employed rigorous experimental methods and provided substantial evidence to support their claims. The use of both simulation and hardware experiments enhances the robustness of the findings. The inclusion of detailed figures and tables, such as the learning curves for single- and multi-critic approaches, as well as the quantitative comparison of goal anticipation policies, aids in comprehending the results effectively.

Clarity

The paper is written with a high degree of clarity. Each section logically flows into the next, providing a coherent narrative from the introduction of the problem to the presentation of results and discussions. The explanations of complex concepts, such as the transformer-based encoder and the multi-critic reinforcement learning (RL) approach, are articulated well.

Originality

The originality of this work is notable. The application of transformer-based models to handle sequences of keyframes in robotic control is innovative and demonstrates a creative extension of techniques traditionally used in natural language processing. Furthermore, the multi-critic RL approach to handle varying reward sparsities presents a novel contribution to the field of reinforcement learning and robotic control.

Significance

The significance of this work is high. It addresses key challenges in the control of quadruped robots, particularly in dynamic environments with varying keyframe targets. The demonstrated improvements in policy learning efficiency and goal accuracy can have substantial implications for the development of more autonomous and adaptable robotic systems. The successful deployment of these methods on hardware, such as the Unitree Go2 robot, underscores the practical relevance of the research.

**Quality Of The Limitations Section:**

3

**Questions For Rebuttal:**

Reward Weights and Sparsity:
- How sensitive is the performance of the proposed approach to the choice of weights  w_s  and  w_d  in the reward function? What guidelines can you provide for tuning these parameters?

Scalability and Generalization:
- How well do you expect the proposed methods to scale to different types of robots or environments? Have you considered testing the framework on robots with significantly different morphologies or in more complex environments?

Data Efficiency:
- The paper mentions the efficiency of policy learning. Can you provide more details on the data efficiency of your approach compared to other state-of-the-art methods?

Public Availability:
- Is there any plan to make the code and datasets used in this research publicly available? This would greatly benefit the research community and facilitate further advancements in the field.

**Robotics Focus:**

4

**Summary Of Paper:**

To implement keyframe-based control of quadrupeds, the paper proposes to train multiple critics to separately handle sparse and dense rewards. A transformer-based keyframe encoder is used. The method is shown to work better compared to having a single critic or a policy only aware of the next goal (instead of anticipating future goals). Great explanatory video is provided.

**Summary Of Recommendation:**

Solid clean paper with a clear straightforward contribution and strong demonstration on real hardware

---

### Author Rebuttal · Authors · 2024-08-09

The submitted zip file contains a figure and a video.

Figure R1, shows a comparison of reward curves for two policies: one trained with including the state in the token and the other one without including the state in the token. It demonstrates that including state in the tokens helps the policy learn to achieve higher rewards.

The video shows a quick demonstration of training our framework for a different morphology. It demonstrates that the policy is able to guide the robot to its position targets with reasonable locomotion behavior.

---

### Decision · Program_Chairs · 2024-09-04

**Decision:**

Accept

**Comment:**

The paper presents a keyframe-based learning framework for legged locomotion. Reviewer aL5k is excited about the significance of the proposed work, along with its originality. On the other hand, Reviewer i8Sh questioned that the proposed multi-critic framework is not directly related to the proposed concept. Reviewer fZ8e raised concerns about the reproducibility and hardware evaluation. We would like to encourage the authors to respond to all the reviewer's comments.

The authors provided thorough feedback on the raised concerns, and the reviewers seemed satisfied with the responses. Particularly, the authors communicated well with Reviewers i8Sh and fZ8e. To this end, we recommend accepting this manuscript.